# Analysis of Vertical Stiffness Characteristics Based on Spoke Shape of Non-Pneumatic Tire

**Jongkeun Sim** [1], **Jiyeon Hong** [1], **Insu Cho** [2] **and Jinwook Lee** [2,*]

1   School of Mechanical Engineering, Undergraduate Course, Soongsil University, Seoul 06978, Korea; shimjg95@gmail.com (J.S.); ghdwldus0802@gmail.com (J.H.)
2   School of Mechanical Engineering, Soongsil University, Seoul 06978, Korea; gubook@hotmail.com
*   Correspondence: immanuel@ssu.ac.kr; Tel.: +82-2-820-0929

**Abstract:** Recently, research regarding non-pneumatic tires that are resistant to punctures has been actively conducted, and the spoke structure design of non-pneumatic tires has been found to be a crucial factor. This study aimed to analyze the vertical stiffness characteristics of a non-pneumatic tire based on the shape of the spoke under the application of a vertical load. The three-dimensional model of a commercial non-pneumatic tire was obtained from the manufacturer (Kumho Tire Co., Inc., Gwangju, Korea), and the vertical stiffness characteristics of the three tire models with modified spoke shapes were compared and analyzed based on a reference tire model. Results show that the vertical stiffness of the fillet applied model is most appropriate. Furthermore, the vertical stiffness characteristics of the analyzed tire models indicate that if fillets with a minimum size are applied to the spokes, the stability of the non-pneumatic tire is expected to improve.

**Keywords:** airless tire; non-pneumatic tire; spoke shape; branch-shaped spoke; vertical stiffness





## 1. Introduction

The role of a vehicle tire is to support the load of a vehicle, transfer the power generated from the engine to the road surface, change the direction of the vehicle, and act as buffers against vibrations and shocks generated on the road surface. For a tire to function effectively, it is critical that punctures do not occur [1].

Recently, owing to the development of eco-friendly futuristic vehicles, tires have evolved into high-performance tires, and various studies have been conducted to prevent driver anxiety and accidents due to punctures or insufficient air pressure in tires. In particular, because self-driving electric vehicles must be autonomously driven, the limitations of pneumatic tires, which are susceptible to punctures, must be overcome. Accordingly, next-generation tires, such as run-flat, sealant, and non-pneumatic tires as shown in Figure 1, are being developed. However, run-flat and sealant tires cannot fundamentally prevent punctures and can only delay and mitigate the effects of punctures [1]. Therefore, research on non-pneumatic tires, which are resistant to punctures, has been actively conducted recently. Non-pneumatic tires are composed of a shear beam, hub, spoke, and tread. The shear beam functions similar to the belt in pneumatic tires. The hub connects the shaft and tire. The spoke acts in a similar manner as air pressure to support vehicle load, and the tread is grounded with a thick rubber layer. Among them, the spoke structure, which is the most important element, supports the vehicle's load and distributes the impact. Thus, spokes can replace the carcass, sidewalls, and inner liners of pneumatic tires. Hence, many studies regarding the structural design of spokes have been conducted.

S. P. Jung et al. analyzed the mechanical properties of honeycomb and straight spokes of non-pneumatic tires according to their shape [2]. A.M. Aboul-Yazid et al. compared and analyzed the mass, stress, and rolling resistance of non-pneumatic tire models, developed by Michelin, Resilient Technologies, and Bridgestone [3]. F. Meng et al. analyzed the performance of non-pneumatic tires by spoke type and designed a new spoke shape with a

rectangular structure [4]. X. Jin et al. investigated the dynamic and static behaviors of a non-pneumatic tire with honeycomb-structured spokes according to the angle of spokes [5]. G. Jang et al. performed topology optimization to design an optimal non-pneumatic tire spoke pattern with a static stiffness similar to that of a pneumatic tire and subsequently proved the validity of the topology optimization [6]. In addition, various analytical studies on non-pneumatic tires have been conducted based on the spoke shape design and finite element analysis [7–16].

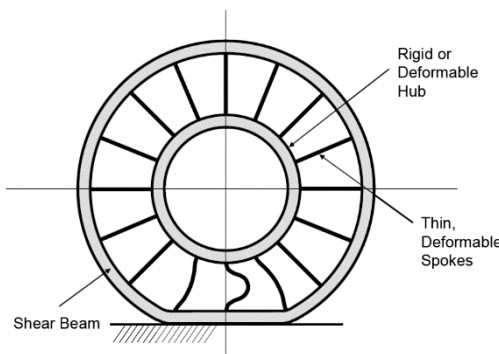

**Figure 1.** Main components of non-pneumatic tire [3].

In this study, a numerical analysis was performed using the non-pneumatic tire model provided by Kumho Tire Co., Inc. (Gwangju, Korea). The tree branch structure of the spoke was kept constant, and the analysis variables were set as fillets applied to the spoke shape, asymmetric spoke division, and symmetric spoke division. A numerical analysis was performed using ANSYS software to compare and analyze the vertical stiffness characteristics [17] based on the shape changes of the deformed structure and existing model.

## 2. Numerical Model and Analysis Conditions of Non-Pneumatic Tire

### 2.1. Development of Finite Element Analysis Model

2.1.1. Base Model of Non-Pneumatic Tire

Tires are essential components of vehicles as they are directly associated with safety. Furthermore, tires can be regarded as a "technological conglomerate" that has been developed through various research and development projects. Therefore, a tire can be designed accurately using a significant amount of information and technical experience. A three-dimensional (3D) model of a non-pneumatic tire provided by Kumho Tire Co., Inc., as shown in Figure 2, was used as the reference model in this study. This model included a straight-type spoke, but it was confirmed that it did not completely reflect the shape of a tree-branch structure.

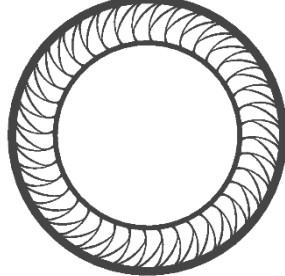
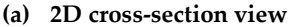
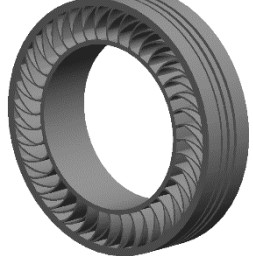

**(a) 2D cross-section view**          **(b) 3D cross-section view**

**Figure 2.** Non-pneumatic tire 3D model used in this study.

Figure 3 shows a detailed drawing of the model (unit: millimeters). The inner diameter

of the tire was 355.60 mm, which is comparable or equivalent to 14-inch wheels. Light-duty vehicles, such as KIA RAY and GM Chevrolet Spark, use this type of tire. The outer diameter and width of the tire were 554 mm and 130 mm, respectively. The thickness of the shear beam was 17 mm, and the thickness of the inner band in contact with the wheel was 10 mm. There are three rows of grooves on the outer surface of the tire, which represent a simplified model of the tread. The illustration on the left in Figure 3 shows the spokes of the tree branch structure at a scale of 2:1.

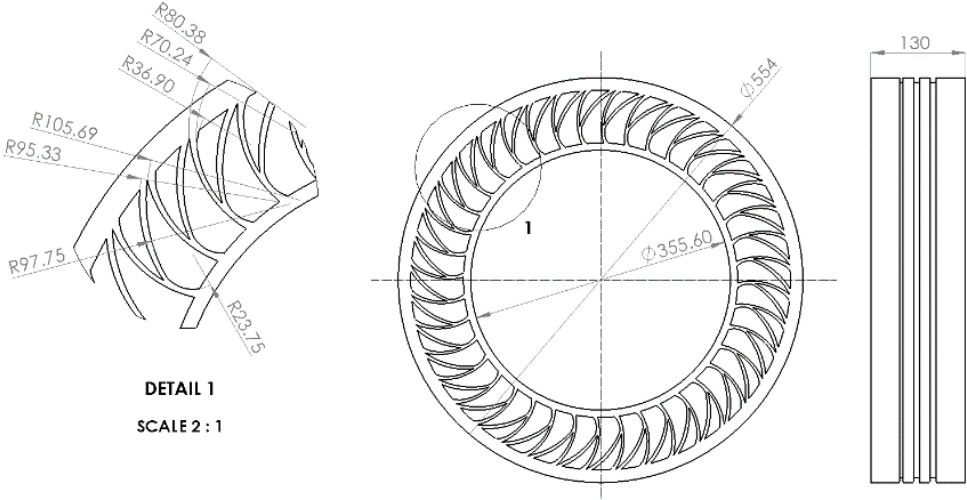

**Figure 3.** Detailed dimensions of non-pneumatic tire.

Between the two spokes, the radius of the outer arc of the long part was 80.38 mm, and the radius of the inner arc was 70.24 mm. In addition, the arc was refracted once such that it was in contact with the shear beam, and the radius of this arc was 36.90 mm. Between the two spokes, the radius of the outer arc of the short part was 105.69 mm, and the radius was then refracted to 97.75 mm. The arc of the part in contact with the inner band had a radius of 23.75 mm, and the radius of the inner arc of the short part was 95.33 mm. The tree branch structure comprised 40 spokes, and the optimized results of numerous studies conducted by Kumho Tire Co., Inc. were used as the detailed dimensions.

### 2.1.2. Modeling of Spoke Deformation of Non-Pneumatic Tire

The 'SolidWorks', as an analytic program tool, was used to modify the spoke shape of the non-pneumatic tire model. To analyze the vertical stiffness characteristics according to the deformation of the spoke shape, the outer and inner diameters, the thicknesses of the shear beam and inner band, and the width of the tire were not modified. Only the shape of the spoke was modified.

The tree branch structure of the spoke model was maintained, and the overall shape was modified into three types: the fillet-applied model, asymmetric-spoke division model, and symmetric-spoke division model.

(1)    Fillet-applied model of spoke

Figure 4 shows the overall shape of the fillet-applied model, in which a fillet is applied to the part of the spoke that experiences concentrated stress and tensile stress. A detailed view of the spokes is also presented in the figure.

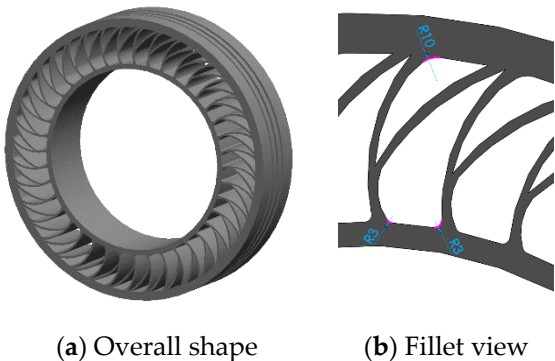

(**a**) Overall shape      (**b**) Fillet view

**Figure 4.** Fillet-applied model.

The fillet radii were set as 10 mm and 3 mm for the shear beam and inner band connections, respectively. Excessive application of the fillet can increase the weight of the tire and lead to a large stress. Therefore, the model was modified by applying the minimum dimensions that preserved the design characteristics of the tire developed by Kumho Tire Co., Inc.

(2) Asymmetric-spoke division model

Figure 5a shows the front view and 3D sectional view of the asymmetric-spoke division model, where the width of original (Kumho Tire) model and the shape of the spoke branches were maintained. The model was designed to improve the vertical stiffness by dividing the spoke length in the width direction in half and placing it asymmetrically in both directions. Figure 5b shows a detailed cross-sectional view of the asymmetric-spoke division model, and the length of the spokes in the width direction were designed to be 64.9 mm each. To minimize interference when the spoke deforms and to prevent a decrease in rigidity due to the reduction in size, a minimum clearance value of 0.2 mm was applied as the spoke spacing.

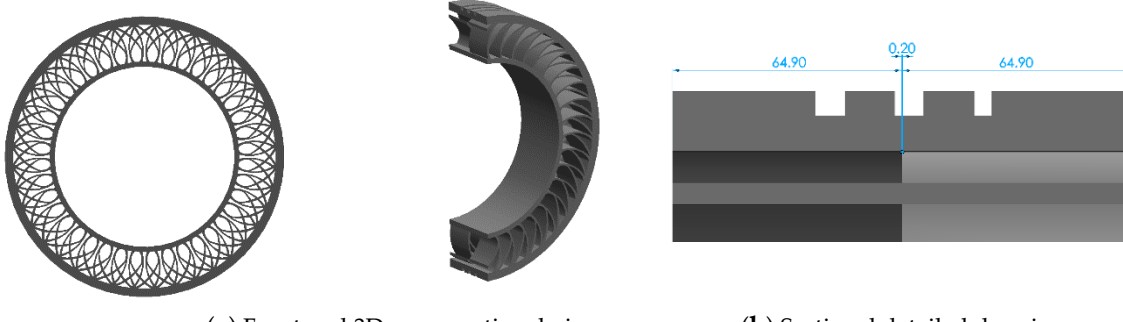

(**a**) Front and 3D cross-sectional views      (**b**) Sectional detailed drawing

**Figure 5.** Asymmetric-spoke division model.

(3) Symmetric-spoke division model

Figure 6a shows a three-dimensional view and a 3D cross-sectional view of a symmetric-spoke division model, where the width of the Kumho Tire model and the basic shape of the spokes were maintained. The model was segmented into three parts and arranged symmetrically by dividing the length of the spokes in the width direction to prevent distortion during loading. Figure 6b presents a detailed cross-sectional view of the symmetric-spoke division model, and the lengths of the two side spokes and the central spoke were designed to be 43.27 mm, respectively, in the width direction. To minimize interference when the spokes deform and to prevent a decrease in rigidity due to size reduction, a clearance of 0.1 mm was applied to the spokes, i.e., the minimum margin value for the gap.

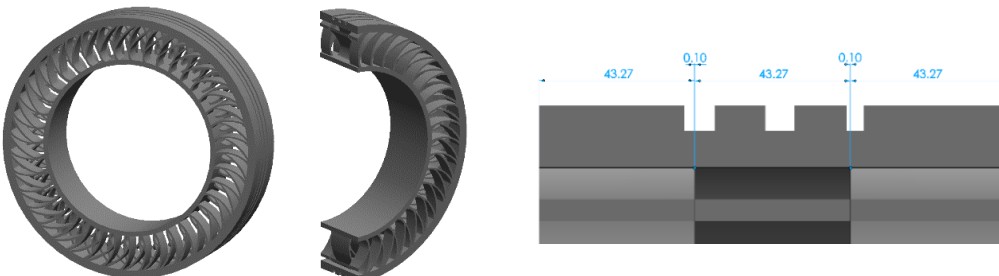

(**a**) Front and 3D cross-sectional views　　　　　(**b**) Sectional detailed drawing

**Figure 6.** Symmetric-spoke division model.

### 2.2. Set-Up of Finite Element Analysis

For an accurate finite element analysis, accurate mesh and geometry settings are essential. In addition, the results may differ significantly based on boundary conditions. Therefore, the correct boundary conditions must be set. Polyurethane was applied as the spoke material based on the data provided by Kumho Tire; the mechanical properties [2,10] of polyurethane used for the finite element analysis are shown in Table 1. Since polyurethane is a rubber-like material, the impact of quasi-static stress-softening behavior and rate-dependent viscous effects should be considered. S. Wang et.al. developed a thermodynamically consistent constitutive model which accounts for both of those phenomena simultaneously [18–20]. In this study, numerical analysis was performed based on a simplified model to understand the characteristics of non-pneumatic tires.

**Table 1.** Material properties of polyurethane [10].

| Properties | Unit | Value |
|---|---|---|
| Density | kg/m$^3$ | 1210 |
| Elastic modulus | MPa | 35.0 |
| Poisson's ratio | - | 0.48 |
| Yield strength | MPa | 145 |
| Shear modulus | MPa | 11.8 |

### 2.2.1. Geometry Set-Up

The design module of ANSYS involves setting up a model for analysis by modeling an object or loading a modeled file. The fillet-applied, asymmetric-spoke division, and symmetric-spoke division type non-pneumatic tire models were created using SolidWorks. To perform the numerical analysis, 400 (length) × 150 (width) × 10 mm (thickness) model of a flat road surface was developed. This surface was set to be in tangential contact with the tire. Figure 7 shows the reference model (among the four non-pneumatic tire geometry models) produced using ANSYS Design-Modeler.

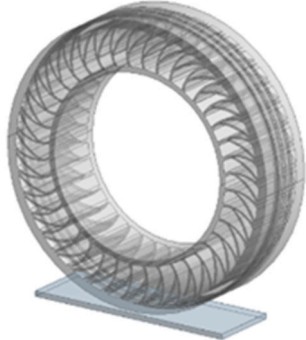

**Figure 7.** Geometry of the reference model.

### 2.2.2. Mesh Set-Up

For the mesh, the "Relevance Center" value was set as "Coarse", and the "Size Function" was set as "Adaptive" to reduce the analysis time for complex shapes, thereby increasing the accuracy of the numerical analysis results. In addition, the appropriateness of the mesh setting was determined using the parameter set system. The mesh size was set as the reference parameter, and the total deformation was set as the parameter value. A simulation was performed using the parameter set, and the results are shown in Figure 8. The result indicate that when the mesh value was less than 0.01 m, the total deformation increased significantly, and an error occurred in the solver at less than 0.006 m because of the complex geometry. In other words, no convergence occurred. Thus, the mesh value was set as 0.008, and an appropriate analysis was performed considering the numerical analysis time. Figure 9 shows the mesh of the reference non-pneumatic tire model, and Table 2 presents the number of nodes and elements of each non-pneumatic tire model under the mesh conditions. The four types of models, i.e., the reference model, fillet-applied model, asymmetric-spoke division model, and symmetric-spoke division model are indicated as Model Types (a), (b), (c), and (d), respectively.

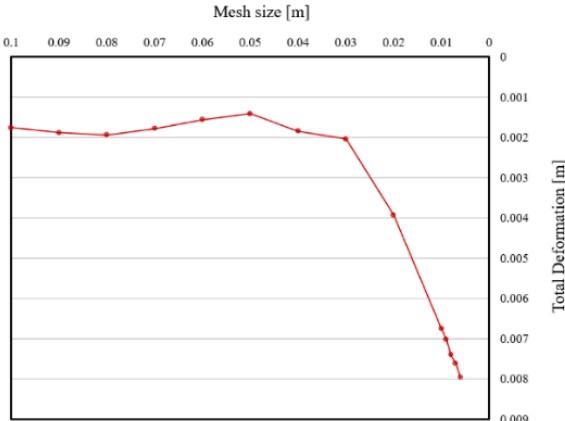

**Figure 8.** Result of parameter set.

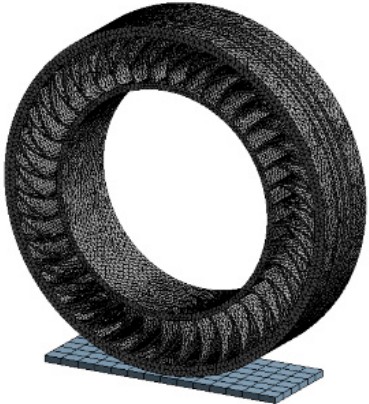

**Figure 9.** Mesh of the reference model.

**Table 2.** Number of nodes and elements in meshes of four non-pneumatic tire models.

| Non-Pneumatic Tire Models | Nodes | Elements |
|---|---|---|
| Model type (a) | 326,781 | 181,737 |
| Model type (b) | 342,969 | 191,189 |
| Model type (c) | 339,744 | 187,092 |
| Model type (d) | 335,785 | 193,463 |

### 2.2.3. Boundary Condition Set-Up

Assuming that the vehicle is stationary, only the vertical stiffness characteristics of the four non-pneumatic tire shapes are considered and analyzed. Therefore, the boundary conditions were set assuming a quasi-static state. Because the inner diameter of the non-pneumatic tire model was 355.60 mm, the conditions were set assuming a 14-inch tire used in light vehicles. Figure 10 shows the boundary conditions of the non-pneumatic tire analysis model.

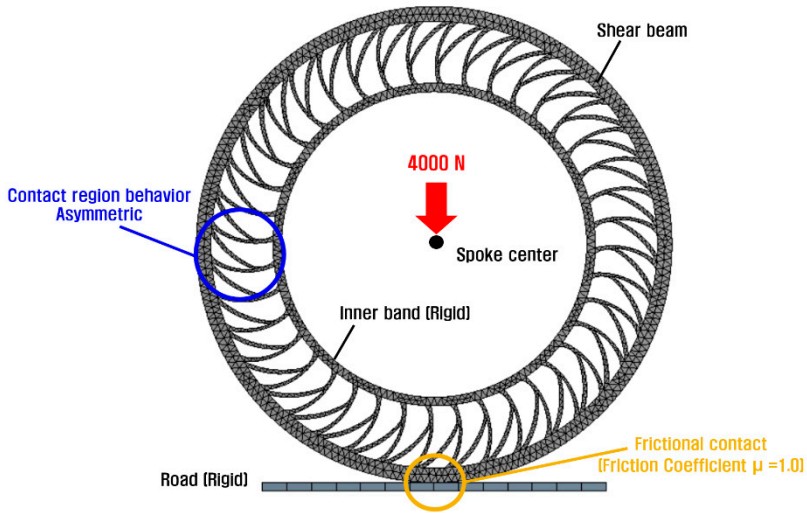

**Figure 10.** Boundary conditions for quasi-static simulation.

Assuming that the tire is connected to the hub of the wheel, the inner band surface was set as a rigid body. "Remote Displacement" is a function of ANSYS that can limit the range of movement of the axe. Using this function, to measure the vertical stiffness, the displacement of the Y-axis was set as "Free", so that the model could freely move upward and downward. In addition, due to the characteristics of the spokes, the structure can undergo tension or compression, and a slight twist or rotation of the tire may occur. Thus, using the aforementioned function, the rotations of the X-, Y-, and Z-axis were set as "Free". Figure 10 shows a representative loading condition, in which a load of 4000 N is applied in the Y-direction perpendicular to the tire. In the analysis, the load is increased from 1000 N to 4500 N in increments of 500 N. The model of the flat road surface was set as a rigid body such that it does not deform when a load is applied. The contact between shear beam and the road was set to frictional contact, and the friction coefficient μ was set to 1.0 by referring to the publication of S. P. Jung et al. [2]. In addition, "Contact region behavior" was set to "Asymmetric" to prevent self-contact between the spokes.

## 3. Results and Discussion

### 3.1. Validation of Non-Pneumatic Tire Analysis Model

The spoke design for the reference model was verified by comparing the vertical stiffness of the 14-inch pneumatic tire and the non-pneumatic tire reference model provided by Kumho Tire Co., Inc.

Based on previous results, it was confirmed that the vertical stiffness of the 175/70 R14 tire was 187,894 N/m when the inflation pressure was 200 kPa, whereas it was 226,415 N/m when the inflation pressure was 250 kPa [18,19].

As shown in Table 3, the average vertical stiffness of Kumho Tire's reference model was 216,745 N/m, which is within the general vertical stiffness range of a 14-inch pneumatic tire. Therefore, Kumho Tire's spoke design was confirmed to be valid, and the non-pneumatic tire model was suitable for analyzing the vertical stiffness characteristics of each spoke shape.

**Table 3.** Analysis results of non-pneumatic tire.

| Force (N) | Directional Deformation (mm) | Vertical Stiffness (N/m) |
|:---:|:---:|:---:|
| 1000 | 4.614 | 216,746 |
| 1500 | 6.924 | 216,744 |
| 2000 | 9.228 | 216,743 |
| 2500 | 11.534 | 216,750 |
| 3000 | 13.841 | 216,747 |
| 3500 | 16.148 | 216,745 |
| 4000 | 18.455 | 216,743 |
| 4500 | 20.762 | 216,742 |
| Average | | 216,745 |

### 3.2. Effect of Change in Spoke Shape on Total Deformation

Table 4 shows the results of the total deformation for each model under a load of 4000 N. In all four models, it was confirmed that the displacement change was the greatest in the lower spoke near the ground, and the maximum deformation occurred at the part where the spoke began to separate into two (red part). The displacement change in the fillet-applied model was the smallest (27.408 mm) whereas it was the largest (29.631 mm) in the symmetric-spoke division model. Compared with the reference model, the total deformation was lower by approximately 1.56% for the fillet-applied model, whereas it was greater by 6.43% for the symmetric-spoke division model.

**Table 4.** Analysis results of non-pneumatic tire models (force = 4000 N).

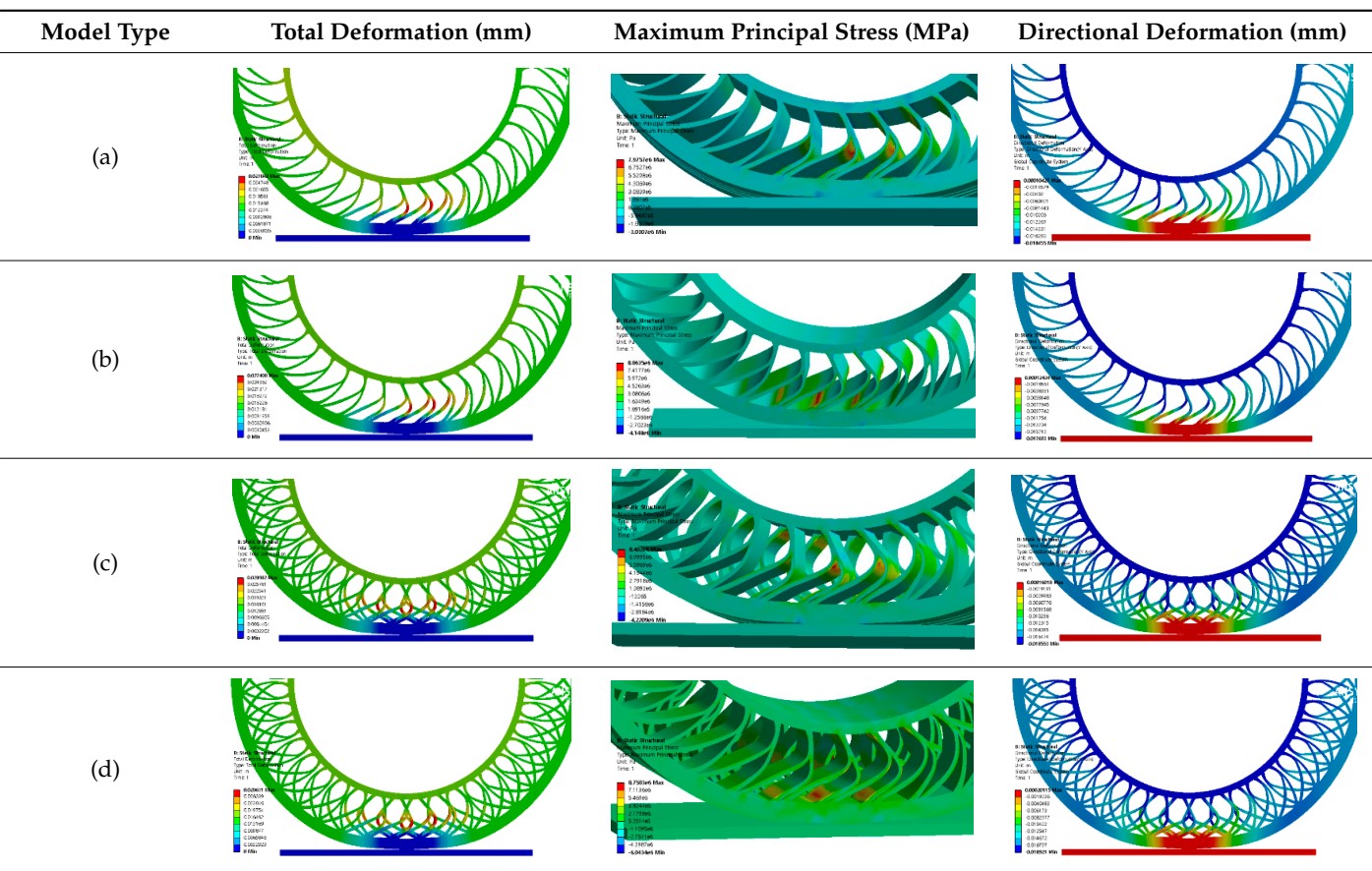

| Model Type | Total Deformation (mm) | Maximum Principal Stress (MPa) | Directional Deformation (mm) |
|:---:|:---:|:---:|:---:|
| (a) | | | |
| (b) | | | |
| (c) | | | |
| (d) | | | |

(Remark) Image resolution cannot be separately downloaded from the analysis program, so individual images are captured and quoted.

### 3.3. Effect of Change in Spoke Shape on Maximum Principal Stress

Table 4 also shows the results of the maximum principal stress analysis for each model under a 4000 N load. In all four models, the load was uniformly distributed over the entire area of the spoke, and the spokes supported the load in a stable manner.

In addition, it was confirmed that the maximum tensile stress occurred due to the bending of the spoke near the shear beam connection among the spokes located near the ground.

The tensile stress was the highest at 8.8635 MPa in the fillet-applied model and the lowest at 7.9757 MPa in the reference model. The compressive stress was the highest at −6.0434 MPa in the symmetric-spokes division model and the lowest at −3.0307 MPa in the reference model.

### 3.4. Effect of Change in Spoke Shape on Directional Deformation

Table 4 shows the results of the directional deformation analysis for each model under a 4000 N load.

In all four models, it was confirmed that a significant change in vertical displacement occurred at the part that was in contact with the ground, and that the spokes bent organically toward each other and supported the load.

In the fillet-applied model, the vertical displacement was the lowest at 17.693 mm, whereas it was the highest in the symmetric-spoke division model at 18.921 mm.

Compared with the reference model, the directional deformation was lower by approximately 4.13% for the fillet-applied model and greater by 2.53% for the symmetric-spoke division model.

### 3.5. Vertical Stiffness Characteristics for Each Spoke Shape

Table 5 presents the vertical stiffness characteristics of each model. The vertical stiffness is defined [3,6] as shown in Equation (1), the load condition is 4000 N, and $\delta_y$ is the result of directional deformation.

$$K = \frac{F}{\delta_y} \tag{1}$$

**Table 5.** Results of static structural analysis.

| Model Type | Maximum Principal Stress (MPa) | Total Deformation (mm) | Vertical Stiffness (N/m) |
|---|---|---|---|
| (a) | 7.976 | 27.842 | 216,743 |
| (b) | 8.864 | 27.408 | 226,078 |
| (c) | 8.402 | 28.982 | 215,599 |
| (d) | 8.758 | 29.631 | 211,405 |

Deformation is an index for determining the vertical stiffness characteristics. As shown in Figure 11a–c, Table 5, and Figure 12, the total deformation of the fillet-applied model was the lowest, and the vertical stiffness characteristics of this model were superior to those of the other models.

For the fillet-applied model, when the spoke was deformed by the load of the tire, the maximum tensile stress occurred above the connection between the shear beam and the spoke.

The application of the fillet interfered with the bending of the spoke, resulting in a greater stress on the spoke compared with the other three models. However, the addition of the fillet hindered bending and decreased the total deformation, thereby reducing the directional deformation, and eventually increasing the vertical stiffness.

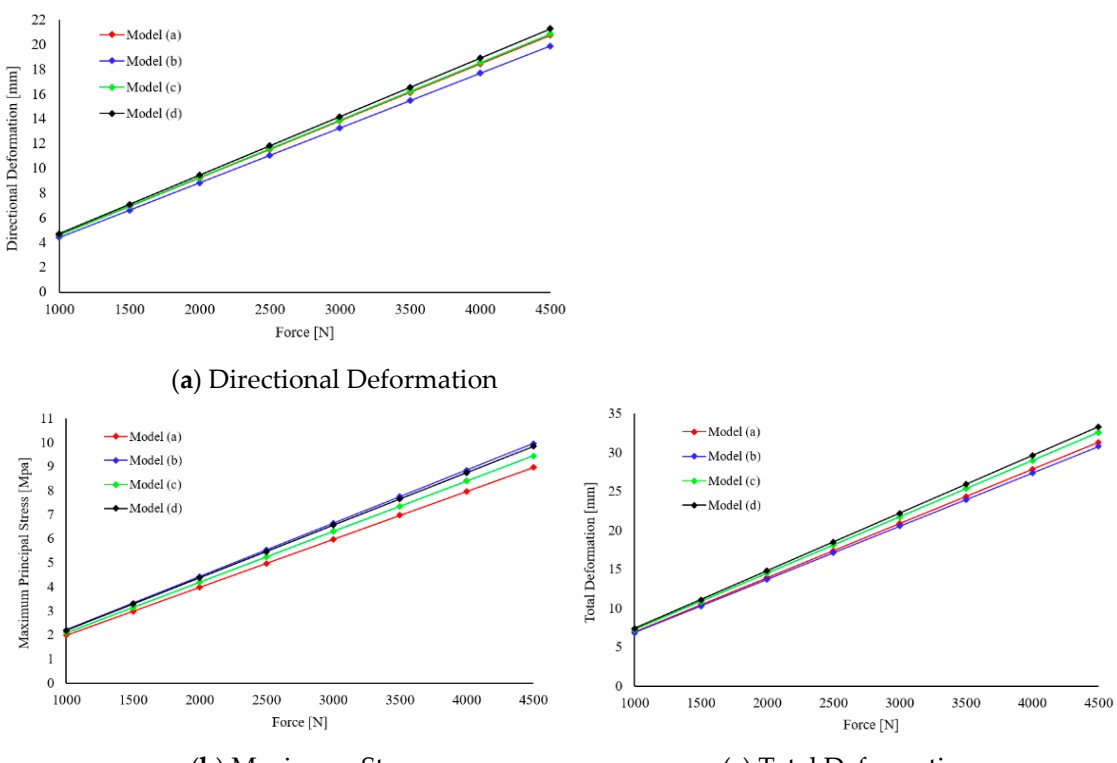

(**a**) Directional Deformation

(**b**) Maximum Stress

(**c**) Total Deformation

**Figure 11.** Static structural analysis results with respect to force.

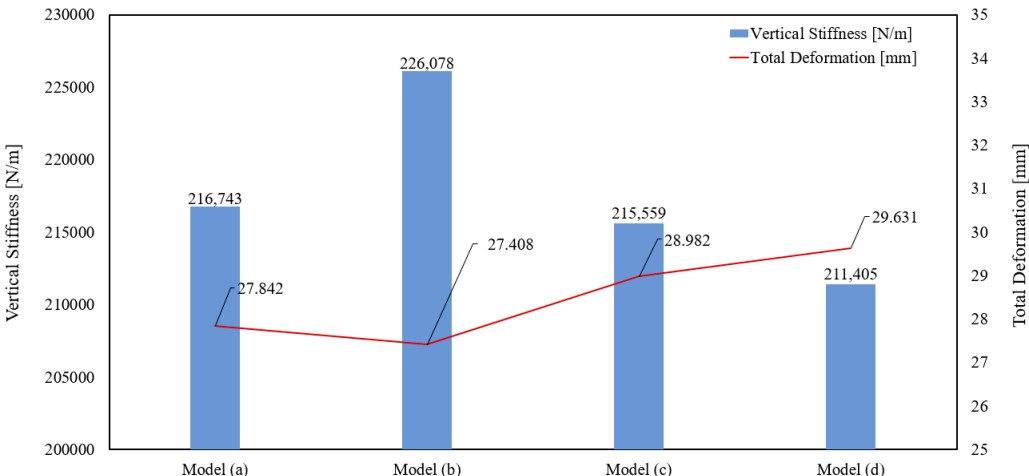

**Figure 12.** Comparison of vertical stiffness and total deformation of four models.

Compared with the reference model and the fillet-applied model, for the asymmetric-spoke-division (Model Type (c)) and the symmetric-spoke-division (Model Type (d)) models, the vertical stiffness was relatively small due to the significant directional deformation. This is attributable to the divided spoke and the weakened resistance to the load in the longitudinal direction.

## 4. Conclusions

In this study, the vertical stiffness characteristics depending on shape change in shape of spokes of the same structure were compared and analyzed via finite element analysis assuming a quasi-static state. The tree branch structure was adopted for the non-pneumatic tire spoke, and a fillet-applied model, asymmetric-spoke division model, and symmetric-spoke division model were used for the analysis. The static analysis results were compared

with the reference model obtained from Kumho Tire Co., Inc., and the vertical stiffness characteristics based on the change in the spoke shape were analyzed. Consequently, the following conclusions were drawn.

(1) Compared with the reference model, the total deformation was lower by approximately 1.56% for the fillet-applied model, higher by 4.09% for the asymmetric-spoke division model, and higher by approximately 6.43% for the symmetric-spoke division model.

(2) Compared with the reference model, the maximum principal stress was higher by approximately 11.13% for the fillet-applied model, 5.34% for the asymmetric-spoke division model, and 9.81% for the symmetric-spoke division model.

(3) Compared with the reference model, the directional deformation was lower by approximately 4.13% for the fillet-applied model, higher by 0.53% for the asymmetric-spoke division model, and higher by 2.53% for the symmetric-spoke division model.

(4) The fillet-applied model with the smallest deformation indicated the highest vertical stiffness at 226,078 N/m, whereas the symmetric-spoke division model with the greatest deformation indicated the lowest vertical stiffness at 211,405 N/m.

(5) When applying the fillet, the maximum principal stress was relatively large but the directional deformation decreased. Consequently, the vertical stiffness of the non-pneumatic tire increased.

(6) When the spoke was divided, the resistance against the load in the longitudinal direction decreased, deformation increased, vertical stiffness decreased, and the stress acting on the spoke increased.

The findings of this study indicate that the vertical rigidity of a non-pneumatic tire can be secured if a fillet with a minimum size, which does not impair the performance of the tire, is applied at the location where concentrated stress occurs in the spoke. This can ultimately improve the operating stability of a non-pneumatic tire.

Based on the analysis results of this study, a detailed validation will be conducted in the future via an experiment using a prototype non-pneumatic tire.

**Author Contributions:** J.S. and J.L. conceived and designed the data analysis, analyzed the data, and wrote the paper; J.H. performed the analysis and produced the analytic data; I.C. researched the paper and proposed the direction; All authors have read and agreed to the published version of the manuscript.

**Funding:** This research received no external funding.

**Institutional Review Board Statement:** Ethical review and approval are not applicable for this study not involving humans or animals.

**Informed Consent Statement:** Informed consent was obtained from all subjects involved in the study.

**Data Availability Statement:** The data presented in this study are available on request from the corresponding author.

**Acknowledgments:** We would like to express our deep gratitude to Kumho Tire Co., Inc. for supporting this research.

**Conflicts of Interest:** The authors declare no conflict of interest.

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
