# Peer review of "Analysis of Vertical Stiffness Characteristics Based on Spoke Shape of Non-Pneumatic Tire"

_applsci, doi:10.3390/app11052369_

Round 1

Reviewer 1 Report

Content

  • “safety tire” is not a well-defined or commonly-used expression. Google it. If you intend to coin the term, then you need to explicitly define “safety tire”. If it is a literal translation from another language, then you need a better translation.
  • “Non-pneumatic tires are composed of a shear band that functions similar to the belt of a pneumatic tire, a hub that combines the shaft and tire, a spoke that contributes to the pneumatic pressure, a tread, and a thick rubber layer that connects to the road surface.” I do not understand what the three phrases in italics mean.
  • Why are you writing about “pneumatic pressure” of non-pneumatic tires? Perhaps you mean “contact patch pressure distribution”?
  • How do spokes “replace the roll” of the “inner liner”
  • I cannot guess what this means: “the outer surface of the tire comprised three rows of grooves, which were assumed to be used for joining the tread and spoke.”
  • What does “and it was transformed using the Solidworks program” mean? “Transformed” from what into what?
  • What does “were excluded in the non-deformed part” mean?
  • After that, I couldn’t really follow what the authors were trying to say.

Formatting

  • Figure 2 is too small to read, and the gray background does not help. The dimensions are illegible.
  • In Figure 6, it is impossible to see the difference between the 4 tires. Either make the difference clearer or leave it out. It adds nothing to the article as is. Figure 8 is even worse.
  • The detailed conclusion would be much easier to see and comprehend in a table, with many fewer words, compared to the current prose.
  • The abbreviation for megapascals is “MPa”, not “Mpa”. See https://en.wikipedia.org/wiki/Pascal_(unit) “In engineering the megapascal (MPa) is the preferred unit for these uses, because the pascal represents a very small quantity.”

English Language

The English Language is generally pretty good, but a little off for a native speaker. Below are just a few examples. This is not a comprehensive list and fixing only these examples is insufficient.

  • “However, run-flat and sealant tires cannot fundamentally prevent punctures, and they can only delay and improve punctures.” No one would say “improve a puncture”. Perhaps you mean “mitigate the effects of a puncture”.
  • It is odd to say “it has been discovered that punctures can be avoided at the source”. This has been known since the very first pneumatic tires. Eliminating the pneumatic part avoids punctures at the source.
  • “Tire is an essential component …” should probably be “Tires are essential components…”
  • “355.60 mm, which is applicable to 14-inch wheels.” should probably be “355.60 mm, which is comparable or equivalent to 14-inch wheels.”
  • “vehicles that can apply this type of tire” should probably be “vehicles that use this type of tire” or “vehicles to which this type of tires apply”. Vehicles do not “apply” tires.

Author Response

We are very appreciate for your precious reviewing.
In the revised paper, the English part of the correction is written in blue, and the parts reflecting the main reviewing opinions are marked in red and the answer is as follows. Thanks for your valuable comments with greatest care. 

Reviewer 2 Report

Please see the attached PDF.

Author Response

(The authors gave the same response as above.)

Round 2

Reviewer 2 Report

The authors have addressed my comments. The manuscript could be accepted in the current form.